# Sexual Health in Menopause

**DOI:** 10.3390/medicina55090559

**Published:** 2019-09-02

**Authors:** Irene Scavello, Elisa Maseroli, Vincenza Di Stasi, Linda Vignozzi

**Affiliations:** Andrology, Women’s Endocrinology and Gender Incongruence Unit, Department of Experimental, Clinical, and Biomedical Sciences “Mario Serio”, University of Florence, 50139 Florence, Italy

**Keywords:** female sexual dysfunction, genitourinary syndrome of menopause, hypoactive sexual desire disorder, vulvo-vaginal atrophy, dyspareunia, hormonal therapy

## Abstract

Sexual function worsens with advancing menopause status. The most frequently reported symptoms include low sexual desire (40–55%), poor lubrication (25–30%) and dyspareunia (12–45%), one of the complications of genitourinary syndrome of menopause (GSM). Declining levels of sex steroids (estrogens and androgens) play a major role in the impairment of sexual response; however, psychological and relational changes related with aging and an increase in metabolic and cardiovascular comorbidities should also be taken into account. Although first-line therapeutic strategies for menopause-related sexual dysfunction aim at addressing modifiable factors, many hormonal and non-hormonal, local and systemic treatment options are currently available. Treatment should be individualized, taking into account the severity of symptoms, potential adverse effects and personal preferences.

## 1. Introduction

Human sexuality is multifactorial, depending on the integration of psychological, biological, relational and sociocultural determinants. The concept of multidimensionality of sexuality is all the more true in the female gender. Indeed, the sexual response in women is particularly complex, due to an elaborate interplay of neuroemotional responses, pursuit of relational intimacy and dramatic fluctuations in hormonal levels [1].

Sexual health is “a state of physical, emotional, mental and social wellbeing related to sexuality; not merely the absence of dysfunction or infirmity”, and female sexuality has been recognized by the World Health Organization not only as an important component of women’s health, but also as a basic human right [2]. The gradual extension of the life span, with expectancy reaching 81 years in the United States, has resulted in a woman spending on average one third of her life in the postmenopausal stage. Although there is a tendency to assume that women lose interest in sex after menopause, sexuality remains a moderately or extremely important element for many midlife women [3]. On the other hand, a very small percentage of midlife women report having discussed their sexual problems in a medical setting as opposed to men [4]. In fact, additional barriers exist for aging women to access information and professional help with regard to sexuality, with the result that adequate treatment is only rarely sought and/or received. Unfortunately, these barriers frequently involve inadequacy and faults on the part of health care providers, including lack of time, scarcity of specific knowledge, concerns about one’s own confidence and abilities, worry about causing offense, personal discomfort and stereotypes about the lack of sexual needs and age-appropriate behaviours [5].

## 2. Menopause and Sexual Function

### 2.1. Prevalence, Characteristics and Risk Factors for Sexual Symptoms

It is widely accepted that sexual function worsens with advancing menopause status, independently of age. The most frequently reported symptoms include low sexual desire (40–55%), poor lubrication (25–30%) and dyspareunia (12–45%), one of the complications of genitourinary syndrome of menopause (GSM) [4]. Sexual dysfunction (SD) in this time of life is rooted in a wide range of predisposing, precipitating and maintaining factors, which may be of biological, psychological and socio-cultural origin [6] (Table 1). It follows that a multidimensional approach to the management of SD is of particular importance in menopause.

Several surveys have documented the prevalence of sexual complaints in menopausal women across different continents in the last two decades [7,8,9]. In a prospective observational study among more than 430 Australian women (aged 45–55 years), significant changes in sexual function were observed, including decreased sexual responsiveness, sexual frequency, increased vaginal dryness and partner problems, with a significant decline with age and menopause [7]. The prevalence of sexual dysfunction was in the range of 42–88% through the menopause transition [7]. In the Massachusetts Women’s Health Survey II, postmenopausal women reported less sexual desire compared to premenopausal women; similarly, both peri-and postmenopausal women reported lower arousal compared to themselves in their 40s [8]. Finally, data from a European survey conducted with 1805 postmenopausal women (aged 50–60 years) revealed that 34% complained of reduced sex drive and 53% had become less interested in sex, whereas 71% reported that maintaining a sex life was important [9].

It has to be underlined that the presence of low desire, the most common sexual problem in women at midlife, does not represent a SD *per se*; in fact, it is mandatory to evaluate the presence of sexually-related distress in order to diagnose hypoactive sexual desire disorder (HSDD), characterized by persistently low or absent desire that causes significant distress [10]. In this regard, a recent cross-sectional community-based study of 2020 Australian women (40–65 years) found that the prevalence of low desire was 69.3%, whereas that that of HSDD was 32.2% [11]. Several modifiable factors resulted associated with HSDD in midlife women, specifically vaginal dryness (OR 2.08), pain during or after intercourse (OR 1.6), moderate to severe depressive symptoms (OR 2.69) and use of psychotropic medications (OR 1.42) [11].

Although population studies suggest that women often experience a decline in sexual functioning with aging and the menopausal transition, this does not apply to the whole female population; furthermore, how psychosocial aspects and the hormonal environment contribute to the maintenance of a healthy sexual life in menopause has to be clarified.

### 2.2. Hormonal Changes and Sexual Behavior: Beyond Estrogen Deficiency

Sex steroids play a major role in positively modulating sexual behaviour, mood, emotion and cognition throughout a woman’s lifespan. Declining levels during peri-menopause, and extremely low levels in post-menopause, are associated with detrimental consequences on overall and sexual well-being. With regard to the central nervous system, menopause-related low sex steroid levels may result in changes in the activation of certain brain areas, representing the neurobiological correlate of menopause-related decrease in sexual arousability [12]. Functional magnetic resonance imaging (fMRI) highlighted a lower activation in post- versus pre-menopausal women in the thalamus, amygdala and anterior cingulate cortex following visual sexual stimulation [13]. Estrogen loss has been suggested to exacerbate the effects of aging on cognitive functions, contributing to brain senescence and neurodegenerative processes [14]. Moreover, estrogens stimulate several neurotransmitter systems in brain regions involved in sexual behaviour, including dopaminergic transmission [12].

In the last years, research on menopause and sexuality basically focused on estrogen deficiency and their local or systemic replacement. However, androgens recently came back into the spotlight as pivotal regulators of female sexual health. Indeed, preclinical and clinical studies have been constantly indicating that estrogens and androgens exert a synergic, stimulatory effect on the female sexual response, promoting sexual desire through a complex network of neurotransmitters and balance between excitatory and inhibitory signals [1]. It is worth noting that testosterone circulates at higher concentrations than estradiol during both pre- and postmenopausal years [15].

Androgen decline is aging related rather than menopause related; the production of the prohormone of testosterone, dehydroepiandrosterone (DHEA), by the adrenal glands progressively declines, and despite a high production in the ovaries, this is not able to correct the deficit. Conversely, an abrupt drop in testosterone levels is observed in young women undergoing surgical menopause, with removal of the ovarian component of androgen production. Indeed, it has been established that surgical menopause often leads to more distressing symptoms, especially sexual dysfunction, compared with natural menopause, with double the risk of developing low sexual desire [16]. Since bilateral oophorectomy is associated with lower total and free testosterone levels than natural menopause, the reduction in androgens is the most plausible cause of increased risk for HSDD in these women [17]. Many studies demonstrated that treatment with testosterone is effective in improving low sexual desire in both naturally and surgically postmenopausal women, either alone or in combination with hormone replacement therapy (HRT) [10]. Although the molecular mechanisms have not been clarified, it has been proposed that androgens stimulate sexual desire acting on the brain reward systems, probably through a modulation of dopaminergic pathways [18,19].

### 2.3. Genitourinary Syndrome of Menopause (GSM)

Genitourinary syndrome of menopause (GSM) is a definition introduced in 2014 by the International Society for the Study of Women’s Sexual Health (ISSWSH) and the North American Menopause Society (NAMS) [20]. GSM replaced older diagnoses such as “atrophic vaginitis”, a reductive and limiting term which was found insufficient to define the complexity of menopause-associated signs and symptoms and their endocrinological impact [20]. GSM is a chronic, progressive condition which encompasses physiological and anatomical alterations that affect the labia majora/minora, vestibule/introitus, clitoris, vagina and the lower urinary tract tissue, as a result of the decrease in sex hormones levels [20]. While vasomotor symptoms (VMS) generally improve over time, vaginal symptoms usually worsen and do not change without treatment. GSM affects approximately 50% of middle-aged and elderly women [21] and has been reported to exert detrimental effects on body image, interpersonal relations, sexual health and overall quality of life [22]. Women may not be aware of safe and simple specific treatments and rarely seek medical help [21].

Sex steroids are critical modulators of the development and maintenance of a healthy genital tissue. Their deficiency results in genitourinary organs returning to the structure and function more representative of prepuberty. Despite the longstanding perspective that low estrogen levels after menopause can lead to GSM symptoms and signs, several lines of evidence from clinical and preclinical studies suggest that androgens also are important in human genitourinary physiology [23]. Indeed, recent studies have shown that the androgenic receptor is already expressed in structures deriving from the urogenital sinus, including the vagina, starting from the first weeks of gestation [24].

Clinical manifestations of GSM consist of decreased turgor and elasticity of the vagina, shrinking of labia minora, loss of vaginal rugae, pallor, erythema, increased vaginal friability with ecchymosis and petechiae [25]. Urological symptomatology is also present due to the common embryologic origin of the vagina and urethral/bladder tissue form the urogenital sinus [24]. Urological symptoms are defined by the term “lower urinary tract symptoms” (LUTS) and include urgency, frequency, urinary incontinence, dysuria, nocturia and recurrent urinary tract infections (UTIs) [25]. Women affected by GSM often report dryness, decreased lubrication, discomfort or pain with sexual activity, post-coital bleeding, irritation/burning/itching of the vulva and/or vagina and pelvic pain [20].

Many predisposing and/or maintaining factors for GSM have been identified: bilateral oophorectomy, premature ovarian failure, other causes of hypoestrogenism (e.g., postpartum period, hypothalamic amenorrhea, ultralow-dose oral contraceptives), smoking, alcohol abuse, decreased sexual frequency or sexual inactivity, lack of vaginal birth, and cancer treatments including pelvic irradiation, chemotherapy, estrogen receptor modulator therapy and aromatase inhibitors [26]. Differential diagnosis of GSM includes dermatological conditions affecting the vulva (e.g., lichen sclerosus or planus, eczema, dermatitis, chronic vulvovaginitis), vulvodynia, vaginismus, autoimmune disorders, malignancy-local or metastatic, chronic pelvic pain, trauma, foreign bodies and diabetes mellitus-related alterations [26].

### 2.4. Psychological and Relational Predictors for Sexual Dysfunction in Menopause

The Women 40+ Healthy Aging Study showed that in a sample of 93 healthy, sexually active women (aged 40–73 years), psychological and relational parameters including self-esteem, optimism, satisfaction with the relationship and emotional support were able to significantly predict overall sexual functioning and specific sexual domains (i.e., arousal, contentment, orgasm, and pain) [27]. On the contrary, the study revealed no associations between psychosocial factors and desire or lubrication [27]. It is noteworthy that sex steroid alterations were not predictive for sexual functioning in this sample [27]. These results confirm previous data suggesting that sexual response and satisfaction are highly dependent on psychosocial aspects related to well-being in menopausal women (see [28] for a review).

Furthermore, it has to be noted that, for both genders, SD impacts not only the symptom bearer, but also the partner, on a sexual, emotional and interpersonal level. In some cases, SD may represent an adaptive reaction to problems in the sexual relationship. This condition, known as “couple SD”, is particularly common in midlife and beyond, due to the higher risk of sexual symptoms which both men and women face with aging. In a survey of more than 2300 Italian men, a significant correlation was observed between perceived moderate-severe low desire in the female partner and severe erectile dysfunction (ED), and menopausal symptoms were among the factors that men thought contributed to their partner’s lack of desire [29]. In a survey of New Zealand women (mean age 53 years) whose partners had ED, 50% (*n* = 48) reported sexual problems; of these, 14 attributed their sexual problems exclusively to their partner’s ED, 5 reported low sexual desire resulting from their partner’s ED, and 1 reported pain resulting from the poor-quality of her partner’s erection [30]. In the same study, many women experienced improved sexual function when their partner’s ED was successfully treated [30]. In this regard, Jannini and Nappi recently introduced the concept of “couplepause”, in order to highlight the importance of a couple-oriented approach that addresses the sexual health needs of the aging couple as a whole, rather than treating the individual [31].

### 2.5. Associated Morbidities: Cardiovascular and Metabolic Diseases

Genital vascular impairment as a determinant of female SD is still debated. However, as in men, the genital sexual response in women is mainly a vascular-dependent event. Based on available evidence, it has been suggested that endothelial dysfunction due to cardiometabolic insults can lead to vascular insufficiency in female genital tissues [32]. Therefore, the decrease in sexual function—in particular, arousal and lubrication—that women experience with menopause and aging could be associated not only with the changing hormonal *milieu*, but also with a growing burden of cardiometabolic alterations [33]. Cardiovascular (CV) disease has become the leading cause of morbidity and mortality also in women [34]; although the higher CV risk in post- vs. pre-menopause is commonly attributed to the decrease in endogenous sex steroids [35], the extent to which menopause and aging relatively contribute to age-related CV risk has not been clarified.

When considering specific diseases, epidemiologic and observational studies consistently demonstrate that the prevalence of SD is higher in women with diabetes mellitus than controls, and in 2012 a meta-analysis highlighted a higher risk for SD in type 1, type 2 and in “any diabetes” vs. controls (OR 2.27, 2.49 and 2.02, respectively) [36]. Notably, although some studies indicate age, duration of disease, metabolic control or complications—e.g., neuropathy—as independently associated with sexual difficulties in female diabetic samples, depression remains the most well-established risk factor for SD in diabetic women [33]. Evidence on the effect of obesity, metabolic syndrome, dyslipidemia and hypertension, whose prevalence significantly increases with menopause and aging, on female sexual health is limited but suggests a negative association [32,33]. In this regard, the Study of Women’s Health Across the Nation (SWAN) found that greater-than-expected weight loss was correlated with an increase in sexual desire at follow-up in midlife subjects [37]. It has been suggested that the gender gap in the correlation between CV and sexual health—which, according to available data, appears milder in women than in men—could underlie an inadequate methodologic approach based on subjective rather than objective outcome measures (e.g., psychometric instruments investigating complex constructs such as “sexual satisfaction”). In this regard, techniques aimed at the objective, direct investigation of the female genital vascular district, including Doppler ultrasound of the genital vessels, are currently being standardized and validated [38].

Finally, it should be noted that hormone levels and vascular health represent two integrated systems, simultaneously acting in the maintenance of female genital tissues’ integrity. Consistently, recent evidence suggests that endogenous testosterone is able to influence body composition and vascular and metabolic function in recent postmenopausal women, and the absence of ovarian testosterone production in recent postmenopausal oophorectomized women seems correlated with deleterious effects on endothelial function [39].

## 3. Treatment Options for Menopause—Related Sexual Dysfunctions

First-line therapeutic strategies for menopause-related SD include education and addressing modifiable factors [10]. Providing information on normal sexual functioning, emphasizing the role of motivation, the importance of adequate sexual stimulation and the influence of age and relationship length often facilitates positive sexual behavioural changes. Common modifiable risk factors, such as mood disorders/use of antidepressants, sedentary life-style, endocrine disorders (hyperprolactinemia, hypo/hyperthyroidism, diabetes mellitus), gynecological and urological infections or diseases should be adequately investigated and addressed [12]. Involving the partner may be helpful in order to modify negative communication patterns, to address partner’s SD or to modify partner’s pressure or demanding behaviour for sex [40]. Pelvic floor physiotherapy and use of dilators and vibrators may be suggested for local symptoms when non-hormonal strategies are needed.

Secondarily, psychological interventions including behaviour therapy, cognitive behaviour therapy (CBT) and mindfulness therapy has been developed to treat HSDD, arousal and orgasmic problems in women, independent of menopausal status. A recent review found three studies of CBT in women with HSDD to be effective compared with wait-list controls [41].

The prevalence and impact of SD and GSM symptoms on postmenopausal women’s quality of life often makes it necessary to initiate a pharmacological treatment. Figure 1 shows a proposed flow chart for the management of SD in menopausal women.

### 3.1. Hormonal Treatment

#### 3.1.1. Hormonal Replacement Therapy (HRT)

HRT is indicated in the treatment of systemic symptoms of menopause, specifically VMS. VMS associated with sleep disruption, fatigue and impaired quality of life are likely to exert negative effects also on sexual function. Furthermore, estrogen therapy may improve sexuality by treating vaginal atrophy, which often leads secondarily to decreased sexual interest, arousal and response. However, a Cochrane review found that HRT with estrogens alone or in combination with progestogens showed only a small-to-moderate benefit when treating overall SD in peri- and postmenopausal women [42]. When considering GSM, data from as early as 2005 from the Women’s Health Initiative (WHI) showed that 74% of women still reported genital symptoms after one year of HRT [43]. These observations were subsequently confirmed in natural [21] and surgical menopause women [44]. Accordingly, a systemic approach is not currently recommended in patients with isolated GSM or for the treatment of SD, including HSDD [45].

#### 3.1.2. Local Estrogen Therapy

For women with GSM and no other menopausal symptoms, local estrogen therapy is recommended and has been the treatment of choice for decades. These preparations are effective in restoring vaginal and urethral epithelium thickness, pH and vaginal microbiota composition. By increasing genital blood flow and reducing dryness and dyspareunia, local estrogen therapy may also improve sexual genital arousal and orgasmic function [46]. The Endocrine Society Guidelines recommend the use of vaginal estrogen for women without a history of hormone-dependent cancer with symptoms of GSM which are not responsive to non-hormonal local therapies [47].

Local estrogen therapies differ in type of compound, posology and route of administration and in most countries include vaginal tablets (natural estrogens), vaginal cream (estriol and promestriene) and vaginal ring (natural estrogens). A typical administration schedule consists of an initiation dose to be taken daily for approximately two weeks, followed by transition to maintenance therapy, to be taken twice a week as long as needed to manage symptoms. The choice among different local estrogen treatments depends on the severity of symptoms (for example, for severe dryness cream can be preferred to tablets) and also on patient’s preference and on the benefit/risk ratio. In fact, one of the major concerns is exposure to systemic estrogen, which is minimized but not eliminated by the local route of administration. In 2015, Santen and colleagues reviewed relevant articles and found that all investigated local estrogen regimens were associated with acute estradiol absorption, with a peak at approximately 8 h and a return to baseline at 12 h [48]. Low-dose vaginal estrogens, arbitrarily defined as the 7.5-μg vaginal ring and 10-μg tablet, increased plasma estradiol levels during chronic administration, but not above the normal postmenopausal range of ≤20 pg/mL; intermediate-dose vaginal estrogens (i.e., 25 μg estradiol or 0.3 mg conjugated equine estrogen) resulted in plasma estradiol levels ≥20 pg/mL.; finally, high-dose vaginal estrogens (50–2000 μg estradiol or 0.625–2.5 mg conjugated equine estrogen) resulted in levels of estrogen in premenopausal range [48]. For these reasons, low-dose regimens should be preferred, in particular when the possible risks overcome the expected benefit. Promestriene is an analogue of estradiol which is minimally absorbed; however, data on its use in patients at high risk (i.e., with a history of hormone-dependent cancer) are scant [49].

A recent Cochrane review evaluated 30 randomized clinical trials (including 6235 postmenopausal women with VVA), comparing different vaginal estrogenic preparations with each other and with placebo [50]. The authors concluded that all local estrogenic compounds improved symptoms of VVA when compared to placebo, without significant differences in efficacy among the different preparations [49]. No difference emerged in terms of adverse effects among different estrogenic preparations and between local estrogens vs. placebo; however, there was low quality evidence that estrogen cream may be associated with an increase in endometrial thickness compared to an estrogenic ring, perhaps due to the higher dose of cream used [50]. In this regard, a study based on 386 endometrial biopsy samples determined that 1-year treatment with 10 mcg estradiol vaginal tablets did not increase the risk for endometrial hyperplasia or carcinoma in postmenopausal women [51]. Accordingly, progesterone use is not recommended when local estrogen preparations are administered [45].

Patients’ preference is a key point to consider, since adherence and compliance are crucial to obtain a significant improvement of GSM symptoms and quality of life. For example, a Swedish survey suggested that women may prefer using disposable applicators with small tablets to deliver local estrogens and seem to value therapy that does not cause smudges/leakage [52].

#### 3.1.3. Ospemifene and Tissue-Selective Estrogen Complex (T-SEC)

Selective estrogen receptor modulators (SERMs) are synthetic nonsteroidal agents that exert a variable agonistic, antagonistic or neutral effect on the estrogenic receptors of the target tissue [53]. This represents an advantage as it prevents adverse events in non-target tissues. Several studies have examined the role of SERMs in the treatment of local symptoms; Raloxifen and Tamoxifen did not demonstrate an estrogenic activity on vaginal epithelium, whereas Lasofoxifen and Ospemifene showed an agonistic effect [53,54].

Ospemifene, to be taken orally (60 mg daily), was the first compound approved in 2013 by the FDA (United States Food and Drug Administration) and in 2014 by the EMA (European Medical Agency) for the treatment of moderate to severe dyspareunia caused by GSM in menopausal women [45]. Ospemifene should be proposed to those women who are not eligible for vaginal estrogen therapy. It exerts a positive effect on the vaginal epithelium and a minimal effect on the endometrial tissue, with stimulating activity on the bone tissue and with antiestrogenic activity in vitro and in vivo in human breast cancer cells [55]. Efficacy and safety were investigated in double-blind phase III trials in which Ospemifene proved effective in reducing vaginal dryness and dyspareunia, evaluated by vaginal pH, maturation index and change in the severity of “the most bothersome symptom” [56]. Symptoms improved in the first 4 weeks and endured for up to 1 year. Ospemifene is well-tolerated; increase in hot flashes is reported as the most frequent adverse effect [56]. Clinical studies suggest a significant neutral or minimum effect on the endometrium, a safe profile on the bone and cardiovascular systems and no stimulation on the breast tissue; however, further long-term studies on endometrial and breast safety are needed [57]. Indeed, Ospemifene is contraindicated in women with undiagnosed abnormal genital bleeding and known or suspected estrogen-dependent neoplasia, including breast cancer undergoing active treatment.

The tissue-selective estrogen complex (T-SEC) is a pairing of conjugated estrogen (CEE) combined with the SERM bazedoxifene (BZA), developed with the aim of improving VMS and vulvovaginal atrophy (VVA) and preventing bone loss [53]. Recent trials have suggested a significant improvement of vulvovaginal symptoms and dyspareunia and an increase in the percentage of surface cells with a reduction of parabasal cells. Oral treatment with CEE/BZA (0.625/20 mg) improves the severity of VVA-associated symptoms in 56% of cases and is able to normalize vaginal histology and pH. This efficacy seems to persist for 2 years [58].

It should be noted that venous thromboembolism and stroke are risks of all systemic estrogen-based therapies, including SERMs.

#### 3.1.4. Androgens: Systemic and Local Treatment

A clinical diagnosis of “androgen deficiency syndrome” in healthy women is currently advised against by the Endocrine Society Guidelines [59]. This is mainly due to a lack of standardized, accurate assays for measuring serum testosterone in women in clinical practice. Furthermore, although large cross-sectional studies have shown associations between specific androgens and self-reported measures of sexual function, circulating levels do not systematically reflect peripheral tissue exposure and sensitivity, which is extremely variable. Nevertheless, systemic testosterone therapy at high physiologic doses has been convincingly shown to improve desire, arousal, vaginal blood flow, orgasm frequency, and sexual satisfaction in surgically and naturally menopausal women, either alone or in combination with HRT [10,12]. Limited data also suggest a beneficial effect of systemic testosterone on vaginal epithelial health and blood flow [60]. In 24-week phase 3 clinical trials in postmenopausal women with HSDD, the 300 mcg/die testosterone patch was effective in improving sexual desire versus placebo, without concerning side effects [61]. Accordingly, the abovementioned Endocrine Society Guidelines suggest a 3–6-month trial of testosterone treatment in postmenopausal women diagnosed with HSDD [59]. In 2019, a meta-analysis of Randomized Clinical Trials (RCTs)comprising 8480 participants found a significant increase in sexual function, including the number of satisfactory sexual events, measures of sexual desire, pleasure, arousal, orgasm, responsiveness and self-image, and a reduction in sexual concerns and distress in postmenopausal women with testosterone treatment vs. placebo or a comparator [62]. A greater risk of developing acne and hair growth was identified, without serious adverse events [62]. Notably, the transdermal route of administration was associated with a neutral lipid profile and is therefore to be preferred [62].

Testosterone is not currently approved for the treatment of HSDD in women in most countries, including Europe and the United States. A 300-mcg/24-h testosterone patch was licensed in 2006 by the European Medicines Agency (EMA) for HSDD in women undergoing surgical menopause receiving concomitant estrogen therapy but was removed from the market in 2012 for commercial reasons [10]. A 1% (10 mg/mL) testosterone cream is currently approved in Australia for the management of symptoms associated with low testosterone in women [10]. Unfortunately, the lack of specifically designed preparations increases the likelihood of inappropriate dosing and virilization. Therefore, when testosterone-based formulations (approved for hypogonadism in men, or in galenic preparations) are used off-label, serum testosterone levels should be measured at 3–6 months to prevent overdosing [59]. Expert opinion reports usually suggest one tenth of the daily dose prescribed to men undergoing replacement therapy for hypogonadism [10,59]. Clinicians should consider the absorption and pharmacokinetics of the available formulations; for example, when using a 2% testosterone gel, approximately one third of the amount delivered at each depression of the canister piston—one half gram of gel, corresponding to 10 mg testosterone—has to be applied in order to administer 300 mcg testosterone. The gel should be rubbed into clean, dry and intact skin, variously to the thighs or lower abdominal/pubic area. Patients should be carefully evaluated for contraindications and adequately informed about the off-label use of the drug. It has to be noted that testosterone is not only “the hormone of sex drive and libido”, but it is also considered important for women’s physical and mental wellbeing [63]. However, long-term safety and other potential beneficial effects on cognition and cardiovascular health have to been confirmed [64].

Concerning other androgens, a recent meta-analysis showed that systemic treatment with DHEA was not effective in improving desire and sexual function in postmenopausal women with normal adrenal function [65]. On the other hand, intravaginal DHEA (prasterone) has been recently approved by the FDA and the EMA for the management of moderate to severe dyspareunia in menopausal women. When applied intravaginally, DHEA is converted not only into estrogens, but also into androgens such as androstenedione, testosterone and dihydrotestosterone (DHT). In placebo-controlled trials, DHEA vaginal ovules inserted daily decreased vaginal pH and improved the vaginal epithelial maturation index and epithelial thickness; at the same time, an improvement in dyspareunia and all domains of sexual function was observed [66]. Preclinical studies have been recently suggesting that the intravaginal conversion of DHEA into DHT could also exert immunomodulatory effects, counteracting menopause-related chronic inflammation of genitourinary tissues; indeed, in vitro DHT treatment of vaginal human smooth muscle cells was able to reduce the expression of several pro-inflammatory genes and markers, working against the maintaining and perpetuating of inflammatory processes [67]. Although serum estradiol and testosterone do not seem to increase after local DHEA use, further studies are needed to examine its safety in patients with a history of hormonal cancer.

### 3.2. Non-Hormonal Treatment

#### 3.2.1. Central Nervous System Agents for HSDD

No central nervous system agents are currently available to treat SD in postmenopausal women. In 2015, the FDA approved flibanserin, a serotonin 1A receptor agonist and serotonin 2A receptor antagonist for acquired, generalized HSDD in premenopausal women only [10]. Previous studies on the antidepressant effects of this agent were interrupted due to poor results; however, an influence on mood and desire was noticed. As highlighted in a recent pooled analysis of three 24-week randomized placebo-controlled trials (VIOLET, DAISY and BEGONIA), flibanserin 100 mg once daily at bedtime determines an improvement in sexual desire and an associated reduction in sexual distress scores [68]. However, safety concerns surrounded the process of approval, due to a clinically and statistically significant increased risk of dizziness, nausea, somnolence and fatigue [69]. These issues, along with the potential for adverse events when the product was taken with commonly used drugs or alcohol, required additional safety studies. Furthermore, the approval of flibanserin was surrounded by an intense general and scientific debate due to its potential social impact. Indeed, detractors of flibanserin advocate a market-driven overmedicalization of low sexual desire in women and claim that the voluntariness of women’s decision concerning the treatment is being deliberately disregarded [70]. According to this view, most women would be driven to take flibanserin because of the social legitimization of their partners’ sexual needs, or to maintain or save a relationship [71]. It must be noted that the ISSWSH, a leading international organization in the field, openly stood up for the clinical meaningfulness of HSDD as a biopsychosocial condition and for the safety and efficacy of flibanserin [72]. As for postmenopausal women suffering from HSDD, the efficacy of flibanserin vs. placebo was demonstrated in 2017 in a randomized, placebo-controlled trial (PLUMERIA) [73]. In this study, the most common adverse events were insomnia (7.7%), somnolence (6.9%) and dizziness (6.4%) [73].

In randomized, placebo-controlled trials, bremelanotide, a melanocortin-receptor-4 agonist self-administered subcutaneously as desired, was also effective in increasing satisfying sexual events/month [74]. It was approved in 2019 by the FDA to treat HSDD in pre-menopausal women only, similarly to flibanserin.

#### 3.2.2. Vaginal Moisturizers and Lubricants

Local estrogen administration may be contraindicated in hormone-dependent cancer survivors and can prove uncomfortable or unwelcome for women who wish to avoid hormone treatment. In these cases, or when only mild to moderate vaginal dryness is present, guidelines recommend vaginal lubricants or moisturizers as first-line treatment [21,45].

Moisturizers have the ability to retain and accumulate water, which is then released locally resulting in increased hydration, thus mimicking physiological vaginal secretions. This provides a quick relief of local symptoms, especially dryness. Moreover, they are known to facilitate cell migration during inflammatory processes, to favor cellular repair and to restore genital tissues’ integrity. Clinical benefits have been reported to last 2–3 days—longer than those of lubricants; therefore, they are particularly beneficial for women who experience a general local discomfort and are not necessarily sexually active [75]. Moisturizers are usually applied regularly every 2–3 days, but the frequency can be increased in the case of more severe atrophy [75]. Hyaluronic acid, a polysaccharide naturally present in the vagina, plays an essential role in the maintenance of the extracellular structure of the epithelium in the case of inflammatory processes and in the preservation of normal local hydration. Hyaluronic acid-based moisturizers are able to determine a quick relief from vaginal burning, itching and irritation due to GSM [75].

A 1994 study comparing moisturizers and local estrogen therapy identified a significant improvement in vaginal dryness and a reduction in pH in both groups; however, moisturizers were not able to modify the maturation index of vaginal epithelium [76]. Similarly, hyaluronic acid vaginal gel applied every 3 days has been associated with an improvement in GSM symptoms comparable to estriol cream, again without a change in pH [77]. A recent study on a hyaluronic-based vaginal gel also identified a trend of improving of vaginal microbiota composition [78].

While moisturizers are intended for chronic maintenance to replace vaginal secretions, lubricants are specifically designed to reduce friction during coitus, increasing comfort and reducing dyspareunia. They are based on oils, glycerin or silicone and usually prescribed as an adjunct to other local or systemic treatments and applied as needed for sexual activity [79].

In 2018, a meta-analysis of sexuality scores in RCTs comparing moisturizers and lubricants to vaginal estrogens [80] indicated an inferior impact of non-hormonal vs. hormonal local treatments on sexuality scores; nevertheless, dyspareunia was significantly improved also with non-hormonal therapies. It is worth noting that the authors concluded that the small number of subjects considered, including breast cancer survivors, was insufficient to provide definitive conclusions [80].

Along with water and plant-based or synthetic polymers, which are needed for water to adhere to the mucosa, moisturizers contain a variety of excipients aimed at providing the appropriate viscosity and preserving pH, which affect the osmolality and acidity of the product [81]. Similarly, excipients contained in lubricants (such as glycols, microbicides and preservatives) influence their osmolality and other important characteristics. In this regard, in vitro studies have been showing that exposure to hyperosmolar lubricants is associated with cytotoxicity, expressed by damage in epithelial cell lines [82]. On the other hand, microbicides can cause inflammation of the mucosa, vulvar irritation, contact dermatitis and alterations in the microbiota [83].

In clinical practice, most women with VVA can obtain a significant benefit from a good-quality, non-hormonal product, with a synergistic effect with hormonal therapies; however, clinicians should be aware of the importance of the physiological values of pH and osmolality of the compound and provide adequate counseling on frequency and length of treatment in order to increase compliance.

#### 3.2.3. Laser Therapies

Carbon dioxide and erbium: YAG laser therapy for GSM is a relatively recent approach based on local stimulation aimed at increasing the production of collagen in order to improve the elasticity and functionality of the vaginal wall. Notably, laser is a treatment option available also for hormonal cancer survivors. However, VVA or GSM are not specifically listed as an indication for treatment [84]. Several studies have been showing a trend toward safe and effective treatment of GSM in the short term (up to 12 weeks); however, most are lacking in randomization, blinding, placebo and comparison groups. In 2014, a study conducted on 30 women aged 33–56 years with symptomatic AVV showed a significant improvement of pelvic floor elasticity (76.6%) and subjective symptomatology (70.0%) after two months [85]. More recently, a multicenter observational study demonstrated the safety and efficacy of laser therapy for the treatment of GSM and stress urinary incontinence [86]. In conclusion, laser use is limited by the lack of data on long-term safety and efficacy and comparing laser therapy with estrogen therapy and control.

#### 3.2.4. Future Options

In 2018 a phase 2 pilot study was conducted to investigate the efficacy of autologous platelet-rich plasma combined with hyaluronic acid for the treatment of VVA in 20 postmenopausal women with a history of breast cancer [87]. All participants showed an improvement in vaginal dryness and dyspareunia, as expressed by the VHI (vaginal health index) score, after 6 months of treatment [87]. These results need to be confirmed.

## 4. Management of SD in Postmenopausal Breast Cancer Survivors

Breast cancer (BCA) is the most frequent oncologic disease in women; its incidence increased after the introduction of imaging screening and continues to grow [88]. Among the negative consequences on personal life and social functioning, SD is highly prevalent in BCA survivors, due to the psychological consequences of the disease, fear of recurrence, body image concerns, fears of fertility loss, fear of rejection by one’s partner and alterations in self sexual esteem. Furthermore, risk-reducing strategies and hormonal therapies such as aromatase inhibitors (AIs) or tamoxifen can significantly decrease circulating estrogen levels and/or alter estrogens’ effects on target organs, causing a significant impact on sexual function [89]. Chemotherapy-related premature ovarian insufficiency can contribute to estrogen and androgen deficiency, leading to negative sexual consequences both in the brain (HSDD) and in the periphery (vaginal dryness and dyspareunia). Accordingly, a recent meta-analysis of 19 studies reported an overall SD prevalence of 73.4% in BCA survivors, with a total female sexual function index (FSFI) score of 19.28 (normal values >26.55) [90]. HSDD (83%), sexual arousal disorder (40%) and dyspareunia (33%) are the most commonly reported SD in this population [91]. It follows that understanding sexual health needs is an important component of BCA comprehensive care.

Unfortunately, there is a lack of safe treatment approaches. In fact, estrogen-based HRT and androgen replacement therapy, which may act on residual BCA cells directly or indirectly through its conversion into estrogens, are contraindicated in all women with a history of hormonal-dependent cancer, independently of the activity of the disease. Therefore, treatment of SD in these high-risk patients is currently based on sex therapy and psychological interventions for general management and on non-hormonal vaginal lubricants/moisturizers, pelvic floor physical therapy and dilators for local symptoms. However, the tolerability and safety of hormonal options are under debate.

With regard to testosterone, preclinical studies have shown antiproliferative effects in some BCA cell lines [92]; on the other hand, the action of the androgen receptor in BCA in humans has not been clarified [93]. It is worth noting that a recent systematic review suggested that the use of systemic transdermal testosterone to treat HSDD in postmenopausal women does not increase the risk of BCA [94]. As for local therapy, in a phase I/II study a 300 or 150 mcg testosterone cream was applied intravaginally for 4 weeks in 21 postmenopausal BCA patients on AIs with symptoms of VVA [95]. The authors found improved signs and symptoms without increasing estradiol or testosterone levels [95]. In a more recent pilot study conducted on a similar sample, the daily use of a 300 mcg vaginal testosterone cream was associated with a significant improvement of all FSFI subscales [96]. These findings were confirmed by Davis et al. in a double-blind, randomized, placebo-controlled trial with vaginal testosterone cream in BCA patients on AIs; after 26 weeks of treatment, the increase in sexual satisfaction score and the reduction in sexual distress and dyspareunia scores were significantly greater in the testosterone group than in the placebo group, without significant changes in serum sex steroid levels [97]. Despite the promising results, specifically designed, long-term trials in women at high risk are warranted to confirm these observations, and androgens are not currently indicated for women with or at high risk for BCA.

Similarly, evidence on local vaginal estrogen therapies in this population is scant. Efficacy on VVA symptoms has been reported with vaginal estriol and estradiol in a few small trials (see [98] for a review). Local estrogens may be considered in women at lower risk for recurrence, with severe GSM symptoms affecting quality of life and who have failed first-line, non-hormonal treatments. Recommendations issued from The American College of Obstetricians and Gynecologists (ACOG) [99], the Endocrine Society [47], and The NAMS [45] cautiously support the use of local hormone treatments in consultation with the woman’s oncologist. In particular, according to the 2018 consensus recommendations from NAMS and ISSWSH, women with estrogen receptor-positive BCA on tamoxifen with persistent, severe GSM symptoms who have failed non-hormonal treatments and with a low risk of recurrence may be candidates for local hormone therapy [100]. Women with the same symptoms on AIs may be advised to switch to tamoxifen; indeed, small transient elevations in serum hormone levels are less concerning with tamoxifene than with AIs [100]. Although the threshold of estradiol concentrations which may raise the risk of disease recurrence is unknown, it is recommended to select a product that results in the lowest maximum and steady-state concentration [100].

Ospemifene has not been studied in women at risk for BCA and is not FDA approved in these patients. Intravaginal DHEA should be used with caution, especially in women with androgen-receptor positive tumors and using either an AI or tamoxifen. Due to the lack of safety data, off-label use of vaginal testosterone is not recommended. Finally, laser therapy may be considered, with appropriate counseling about the lack of long-term safety and efficacy data [100].

In conclusion, further randomized and controlled studies are needed to provide effective and safe therapeutic alternatives for SD to menopausal women with or at high risk for BCA. At the present time, counseling should include a shared decision-making approach, with a careful evaluation of the woman’s perceived need for treatment vs. fears regarding risk; consultation with the oncology team is recommended [100].

## 5. Conclusions

SD in menopausal women is common and characterized by unique determinants and risk factors that go beyond estrogen deficiency. Nevertheless, sexual issues in menopause are underdiagnosed and undertreated. Education of both healthcare professionals and patients is crucial, as there is a general lack of awareness that many options are available—and new are emerging—to maintain and improve genitourinary and general sexual health. Treatment should be individualized, taking into account the severity of symptoms, effect on quality of life, potential adverse effects and personal preferences.

## Figures and Tables

**Figure 1 medicina-55-00559-f001:**
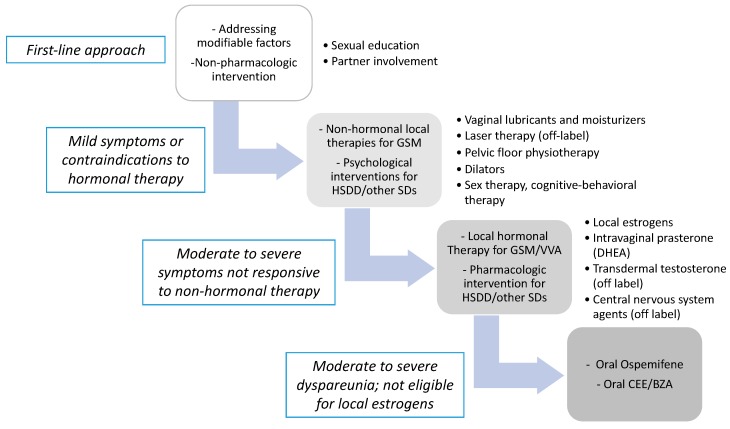
Proposed flow chart for the management of sexual dysfunction in menopause. GSM = genitourinary syndrome of menopause; HSDD = hypoactive sexual desire disorder; SD = sexual dysfunction; VVA = vulvo-vaginal atrophy; DHEA = dehydroepiandrosterone; CEE/BZA = conjugated estrogens/bazedoxifene.

**Table 1 medicina-55-00559-t001:** Factors possibly affecting sexual function in menopause. Adapted from ref [6].

Factors Possibly Affecting Sexual Function in Menopause		
**Predisposing factors**	**Biological**	Gynecological or surgical interventions
		Premature Ovarian Insufficiency (POI)
		Endometriosis
		Iatrogenic menopause (bilateral oophorectomy, chemotherapy, radiotherapy)
		Endocrine factors
	**Psychosexual**	Previous sex life
		Body image
		Personality traits
		History of sexual abuse/violence
		Affective disorders
		Coping strategies
	**Contextual**	Ethnic/cultural/religious expectations and constraints
		Support and network
**Precipitating factors**	**Biological**	Age at menopause (POI)
		Biological vs. iatrogenic menopause
		Iatrogenic menopause
		Extent and severity of menopausal symptoms
		Current disorders
		Substance abuse
	**Psychosexual**	Relationship
		Sexual experience
		Affective disorders
		Loss of partner
	**Contextual**	Life stressors (divorce, separation, partner infidelity)
		Loss or death of close kin
		Lack of access to medical treatment
		Economic difficulties
**Maintaining factors**	**Biological**	Changes secondary to menopause (hormonal, vascular, muscular, neurological, immunological)
		Contraindications to hormone therapy
		Inadequacy of hormone therapy
		Pharmacological treatments
		Substance abuse
	**Psychosexual**	Perception of menopause changes
		Loss of sexual confidence
		Affective disorder
		Distress (personal, emotional, occupational, partner)
		Partner’s general health or sexual problems
	**Contextual**	Lack of access to care
		Interpersonal conflicts

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
