# Peer review of "Sexual Health in Menopause"

_medicina, 2019, doi:10.3390/medicina55090559_

Round 1

Reviewer 1 Report

The authors presents significant topic which is sexual health in menopause. Medical community especially in gynecology should consider to concentrate on this matter more precisely. This manuscript manage to show this problem as a whole. First authors concentrate on the symptoms which may lead us to the right diagnosis. Furthermore the biological causes of the dysfunctions are presented. The work emphasize the influence of steroids hormones on brain function. The authors also include an essential role of chronic diseases in developing SD. Subsequently various ways of treatment are presented. They concentrated on significant role of psychotherapy in managing. Readers may decide by themselves which treatment to choose, by having short information about most of the therapies, their advantages and disadvantages. The authors could explore the topic of non-hormonal treatment more deeply, because a large group of patients will not decide on hormonal treatment, due to concerns associated with it.

From the technical side the manuscript is divided into well understood sections. The graphs are precise and easy to read and might help physicians in remembering risk factors and treatment path. Rich bibliography confirms proper preparation for the subject.

The manuscript is written clear and it is informative, but it does not give full information about the subject. However, it gives an overall look on the problem and leave readers with reflections.

We have two concerns. In our opinion flibanserine as a new treatment option should be described more precisely. Not only as a therapeutic agent but also its social impact which was the subject of doubts by FDA experts. Authors should also include more study about systemic testosterone usage, its dose, and its sexual drive impact. Above concerns seem to be the hot topic of menopausal sexual medicine nowadays.

Author Response

Reviewer #1 Comments and Suggestions for Authors

The authors presents significant topic which is sexual health in menopause. Medical community especially in gynecology should consider to concentrate on this matter more precisely. This manuscript manage to show this problem as a whole. First authors concentrate on the symptoms which may lead us to the right diagnosis. Furthermore the biological causes of the dysfunctions are presented. The work emphasize the influence of steroids hormones on brain function. The authors also include an essential role of chronic diseases in developing SD. Subsequently various ways of treatment are presented. They concentrated on significant role of psychotherapy in managing. Readers may decide by themselves which treatment to choose, by having short information about most of the therapies, their advantages and disadvantages. The authors could explore the topic of non-hormonal treatment more deeply, because a large group of patients will not decide on hormonal treatment, due to concerns associated with it.

 We thank the reviewer for the suggestions. We now added a more detailed review of non-hormonal treatments (moisturizers and lubricants) (please see page 9, paragraph 3.2.2 “Vaginal Moisturizers and Lubricants” – changes highlighted in red).

From the technical side the manuscript is divided into well understood sections. The graphs are precise and easy to read and might help physicians in remembering risk factors and treatment path. Rich bibliography confirms proper preparation for the subject.

 The manuscript is written clear and it is informative, but it does not give full information about the subject. However, it gives an overall look on the problem and leave readers with reflections.

We have two concerns. In our opinion flibanserine as a new treatment option should be described more precisely. Not only as a therapeutic agent but also its social impact which was the subject of doubts by FDA experts.

 We agree with the reviewer on the importance of flibanserin. Our decision to describe only briefly this treatment was due to the fact that it is not approved in menopause. We detailed the discussion including references to the debate relative to its social impact, as suggested (page 8, paragraph “3.2.1 Central nervous system agents for HSDD” - changes highlighted in red).

Authors should also include more study about systemic testosterone usage, its dose, and its sexual drive impact. Above concerns seem to be the hot topic of menopausal sexual medicine nowadays.

We thank the reviewer for this useful comment. The discussion concerning systemic testosterone  therapy was extended accordingly (page 7, from line 347 on - changes highlighted in red).

Reviewer 2 Report

I congratulate the authors about their extensive comprehensive review. They have succeeded in most cases to provide a narrative detailed review about the sexual changes in the menopause. The manuscript is well written and designed .The authors have shown that they have experience in the management of such a crucial social aspect. Needs to.be published.

Author Response

Reviewer #2

I congratulate the authors about their extensive comprehensive review. They have succeeded in most cases to provide a narrative detailed review about the sexual changes in the menopause. The manuscript is well written and designed .The authors have shown that they have experience in the management of such a crucial social aspect. Needs to.be published.

We thank the reviewer for the very positive comment.